
# Non-destructive evaluation of moisture content in wood by using Ground Penetrating Radar

Hamza Reci[1], Tien Chinh Maï[2], Zoubir Mehdi Sbartaï[2], Lara Pajewski[3], Emanuela Kiri[1]

[1]Institute of Geosciences, Energy, Water and Environment, Polytechnic University, Tirana,1024, Albania
[2]I2M Laboratory, Department of Environmental Civil Engineering, University of Bordeaux, 33405, Talence, France
[3]Department of Engineering, Roma Tre University, 00146, Rome, Italy

*Correspondence to*: Lara Pajewski (lara.pajewski@uniroma3.it)

**Abstract.** This paper presents the results of a series of laboratory measurements carried out to study how the Ground Penetrating Radar (GPR) signal is affected by moisture variation in wood material. The effects of the wood fiber direction, with respect to the polarisation of the electromagnetic field, are investigated. The relative permittivity of wood and the amplitude of the electric field received by the radar are measured for different humidity levels, by using the direct-wave method in Wide Angle Radar Reflection configuration, where one GPR antenna is moved while the other is kept in a fixed position. The received signal is recorded for different separations between transmitting and receiving antennas. Direct waves are compared to reflected waves: it is observed that they show a different behaviour when the moisture content varies, due to their different propagation paths.

## 1 Introduction

Ground Penetrating Radar (GPR) is an effective technique that uses electromagnetic waves to obtain three-dimensional images of natural or manmade structures and subsoil. It is employed in a large variety of applications, where non-invasive and non-destructive investigations are required. Examples of applications are surveying of transport infrastructures and buildings, detection and location of utilities, inspection of construction materials, geological and geotechnical investigations, archaeological prospecting and cultural-heritage diagnostics, detection of landmines and unexploded ordnance, planetary exploration and more (Benedetto and Pajewski, 2015; Persico, 2014).

Recently, GPR started being used for the non-destructive evaluation of moisture content in wood material. The most common causes of wood deterioration are biological, due to attacks of fungi and insects, and the moisture content is recognised as the most critical condition for the development of such organisms. For example, it is known that the minimum value of moisture content for the development of wood degrading fungi is 17% by mass, whereas the optimum values range from 30% to 70% depending on the fungi and wood type (Mai *et al.*, 2015). It is then apparent that the non-destructive evaluation of moisture content is of primary importance for the preservation of timber structures.

Few works are present in the literature, concerned with the GPR inspection of wood. In (Lualdi *et al.*, 2003; Muller, 2003; Sbartai, 2011; Martinez *et al.*, 2013b; Mai *et al.*, 2015), the properties of wood were estimated by using reflected-wave





methods. In (Lualdi *et al.*, 2003), GPR was used to detect timber beams and evaluate the type and size of their connection to a bearing wall. In (Muller, 2003), GPR was employed for the inspection of timber bridges, in order to detect piping and rotting defects. In (Martinez *et al.*, 2013b), GPR was used on samples of sawn timber of different species (densities) and interesting results were found: the propagation velocities, as well as the amplitudes of the reflected waves, were always

smaller when the electric field was longitudinal to the grain rather than transverse to it; however, when the field propagated in a random direction, the electromagnetic parameters did not differ significantly. In (Mai *et al.*, 2015), laboratory measurements were carried out with the aim to study the sensitivity of electromagnetic-wave propagation to moisture variation and fiber direction, in Spruce and Pine wood samples. The relative dielectric permittivity was measured by using the resonance technique at 1.26 GHz, and results were compared to GPR measurements carried out with a 1.5 GHz ground-

coupled antenna. The samples were tested in different moisture-content conditions, ranging from 0% to 30% by mass. The real part of the permittivity estimated by using the GPR technique turned out to be in good agreement with that measured by using the resonance technique. The effect of moisture content when the electric field was oriented parallel to fiber direction was observed to be more significant than when the electric field was oriented orthogonal to the fibers.

In (Laurens *et al.*, 2005; Sbartaï *et al.*, 2006a, 2006b; Martinez *et al.*, 2013a), direct-wave methods were used for

the non-destructive evaluation of concrete properties, with successful results. Direct-wave methods are of practical interest, because sometimes it may be difficult to detect the signal reflected by a sample, when applying the technique onsite.

A preliminary example of application of the direct-wave approach to wood assessment can be found in (Maï *et al.*, 2015). In that paper, tests were carried out on a sample of Spruce with humidity of 12%. Measurements were performed with the electric field orthogonal to the wood fibers. The obtained results showed that the direct wave signal is measurable. In

addition, the permittivity values estimated by employing the direct-wave technique turned out to be lower than those estimated from the reflected waves.

Our work focuses on using the GPR direct wave to estimate the properties of wood. Results are compared with those obtained by employing a reflected-wave approach. Different moisture-content conditions are reproduced in the laboratory and analysed. The effects of the wood fiber direction, with respect to the polarisation of the electromagnetic field,

are investigated. An interesting feature of the direct-wave method is that the operator simply has to measure the wave propagating between the transmitter and receiver, without needing a reflector at the bottom of the sample, neither information about the sample thickness.

## 2  Method

The electromagnetic signal received by a GPR in the presence of a wood sample is affected by many parameters such as the

moisture content, wood density, temperature, and direction of fibers (Sahin and Nürgul, 2004; Laurens *et al.*, 2005; Kasal and Tannert, 2010). They influence the electromagnetic-field attenuation, phase shift, and polarisation (Lundegren *et al.*, 2006).





As is well known, electromagnetic waves propagate in the air at a speed of $c = 0.3$ m/ns. In wood, which is a dielectric anisotropic material, the propagation velocity depends on the polarisation of the electromagnetic field. For a fixed polarisation state, the propagation velocity is a function of the relative dielectric permittivity $\varepsilon_r$, the relative magnetic permeability $\mu_r$, the electrical conductivity $\sigma$ and the frequency $f$, as follows (Neal, 2004):

$$v = \frac{c}{\sqrt{\varepsilon_r \mu_r \frac{1+\sqrt{1+(\sigma/\omega\varepsilon)^2}}{2}}} \qquad (1)$$

where $\varepsilon = \varepsilon_r \varepsilon_0$ is the dielectric permittivity of wood, $\varepsilon_0 = 8.854 \cdot 1012$ F/m is the dielectric permittivity in a vacuum, $\omega = 2\pi f$ is the angular frequency and $\sigma/\omega\varepsilon$ is the loss factor. In non-magnetic ($\mu_r = 1$) low-lossy materials as wood, where $\sigma/\omega\varepsilon \approx 0$, the propagation velocity can be approximated by the simple expression $v = c/\sqrt{\varepsilon_r}$.

Several phenomena lead to an attenuation of the electromagnetic signal strength, the most important being the spherical spreading of the energy, which causes the so-called propagation losses, the field reflection and transmission at interfaces between media with different electromagnetic properties, and the scattering. The latter mainly occurs when objects with size similar to the wavelength are present, hence in wood this phenomenon is more pronounced for higher frequencies. The field attenuation due to propagation losses is a function of the dielectric permittivity, the magnetic permeability $\mu = \mu_0 \mu_r$ (where $\mu_0 = 4\pi\ 10^{-7}$ H/m is the magnetic permeability in a vacuum), and the electrical conductivity and the frequency, as follows (Neal, 2004):

$$a = \omega \sqrt{\varepsilon\mu \frac{\sqrt{1+\left(\frac{1}{\omega\varepsilon}\right)^2}-1}{2}} \qquad (2)$$

In low-lossy materials as wood, the attenuation coefficient can be approximated by the simple expression $a = 0.5\sigma\sqrt{\mu/\varepsilon}$ : the attenuation is proportional to the electrical conductivity, which leads to higher attenuation in materials with higher electrical conductivity.

In this work, GPR radargrams are recorded by using two different techniques: direct and reflected waves are measured by using the Wide Angle Radar Reflection (WARR) and Fixed Offset (FO) methods, respectively, as will be explained in the following. Ground-coupled antennas are employed, with central frequency 1.5 GHz; the radar system is a GSSI SIR 3000. A wood sample of Epicea (Spruce) type is used, which is 600 mm long, 190 mm wide and 176 mm thick (see Figure 1). Measurements are carried out in two directions: longitudinal (as in Figure 1.A), where the electric field is polarised orthogonal to the wood fibers, and transversal (as in Figure 1.B), where the electric field is parallel to the fibers. In order to easily distinguish between reflected and direct waves, a metallic sheet is placed under the wood sample.

Measurements starts at a humidity level of the wood sample equal to 12%. Afterwards, the sample is immersed into the water in order to gradually increase its moisture content. GPR experiments are then repeated at different humidity levels. Humidity by mass water (%) is calculated by adopting the following expression (Moron *et al.*, 2016):

$$\text{Humidity (\%)} = 100\left(\frac{W - W_0}{W_0}\right) \qquad (3)$$





where $W_0$, stands for the weight of the sample with 12% humidity, W is the weight of the sample after being immersed into the water. The weight of the sample is measured with a balance having a sensitivity to the gram.

Figure 2 shows the humidity by mass water of the sample, as a function of the time of immersion into the water. As can be seen from the graph, during the first 30 hours the humidity increases sharply; then, an almost curvilinear increase

occurs. The measurements are performed at the humidity levels listed in Table I.

When applying the WARR technique, a radar antenna is kept in a fixed position and the other antenna is moved on the wood surface with a 1-cm step. The distance between the two antennas varies fron 16 to 26 cm and from 11 to 21 cm, for orthogonal and parallel polarisation of the electric field, respectively. When the FO method is applied, the distance between the antennas is 16 and 11 cm for orthogonal and parallel polarisation states, respectively. The arrival times are visualised

with Radan Software and Matlab.  Examples of A-scans showing the superposition of direct air wave, direct wave and reflected wave are provided in Figure 3; here, the polarisation of the electric field is orthogonal to the fibers and the humidity level by mass water is 12%.

For the WARR technique, the propagation velocity is estimated from the arrival times of the direct waves, measured at difference distances between the antennas (the arrival time is the instant corresponding to the first and highest positive

peak in the radargram). In particular, the propagation velocity is estimated as the slope of the linear regression of the arrival time of the direct wave, as a function of the distance between antennas. This is shown in Figure 4, for both polarisation cases and a level of humidity equal to 18.18%.

For the FO method, the propagation velocity v in the wood sample is determined from the peaks of the air wave and reflected wave and the following expressions are used. For the direct air wave, the arrival time is:

$$t_{air} = t_0 + t_{air}^{real} = t_0 + \frac{d_{TR}}{V_0} \qquad (4)$$

where $t_{air}^{real}$, is the arrival time of the air wave (reference signal), $t_0$ is the starting time of electromagnetic impulse and $d_{TR}$ is the distance between the transmitting and receiving antennas. For the reflected wave, the arrival time is:

$$t_r = t_0 + t_r^{real} = t_0 + \frac{d_R}{V} \qquad (5)$$

where $t_r^{real}$ is the arrival time of the reflected wave and $d_R$ is the length of the propagation path of the reflected wave, which

is given by the equation:

$$d_R = 2\sqrt{(\frac{d_{TR}}{2})^2 + h^2} \qquad (6)$$

where $h$ is the thickness of the wood sample. Combining Equations (4) and (5), it is possible to estimate the propagation velocity inside the wood sample, as follows:

$$v = \frac{d_R}{\Delta t + \frac{d_{TR}}{V_0}} \qquad (7)$$

where $\Delta t = t_r^{real} - t_{air}^{real}$. Finally, the relative permittivity of the wood sample can be estimated as $\varepsilon = (c/v)^2$.



## 3 Results and discussion

As mentioned in Section 2, the wood relative permittivity is measured for different humidity levels (ranging from 12% to 64.65%) and polarisation cases (electric field orthogonal and parallel to the wood fibers). Results are summarised in Table I and plotted in Figure 5.

When the direct-wave method is used, the estimated value of the relative permittivity does not significantly change if the polarisation is rotated. When the electric field is parallel to the fibers, the permittivity values are systematically higher than those measured when the electric field is orthogonal to the fibers. The increase of relative permittivity versus moisture content is piecewise linear, with a slope change occurring when the humidity level is about 18%.

      For the reflected-wave method, the increase of relative permittivity versus moisture content is piecewise linear as
well, with a higher slope than in the case of the direct-wave method. Moreover, the slope does strongly depend on the polarisation of the electromagnetic field and this is in agreement with (Martinez *et al.*, 2013b, and Mai *et al.*, 2015). When the electric field is orthogonal to the wood fibers, a slope change occurs at a humidity level of about 18%, corresponding to the fiber saturation point. The slope change is less visible and seem to occur at higher humidity levels, when the electric field is parallel to the wood fibers; this is again in agreement with (Martinez *et al.*, 2013b, and Mai *et al.*, 2015).

At all humidity levels, the permittivity values measured by the reflected-wave method are consistently higher than those measured by the direct-wave method. For both methods, the direction of the fibers does not affect the wood permittivity when the moisture content is low, and then it becomes more important in the presence of higher humidity levels. It is interesting to notice that the results of the reflected-wave method are closer to the direct-wave curves when the electric field is orthogonal to the wood fibers. Apparently, the propagation paths are similar in the two cases.

The obtained results show that direct waves in wood behave differently than reflected waves. This happens because the direct and reflected waves follow different propagation paths: the direct waves propagate in the top layer of the sample and the effect of the electromagnetic-field polarisation is small; the reflected waves propagate through the whole sample and, due to the anisotropy of wood material, the polarisation has a stronger effect on the results.

      When the electric field is orthogonal to the wood fibers, direct waves can be distinguished even when the humidity
levels are above 60%. When the electric field is parallel to the wood fibers, instead, the direct wave arrival time cannot be detected for humidity levels higher than 43%. Indeed, a high dissipation of electromagnetic energy occurs and the waves are highly attenuated.

      A further goal of this work is to study how the distance between the radar antennas affects the amplitude of the received signal. For each considered humidity level, the amplitude of the direct wave is then measured with antennas placed
at different distances. In Figure 6, the direct-wave amplitude normalised to the amplitude of the direct air wave is plotted, as a function of the distance between transmitting and receiving antennas, when the humidity by mass water is 18.18%. As expected, the amplitude shows an exponential attenuation when the distance increases. In Figure 7, the normalised amplitude of the direct wave is plotted as a function of the humidity level, for both parallel and orthogonal polarisation cases, when the





distance between the antennas is 11 cm and 16 cm, respectively. It can be noticed that, when the moisture content increases, the normalised amplitude at small distances turns out to be higher than one, when the electric field is orthogonal to the wood fibers: this may be due to a superposition of direct and direct air waves. For small humidity levels, the normalised amplitude increases with the moisture content; then, when the moisture content is further increased, the normalised amplitude starts to

decrease (this happens at about 30% and 25% humidity by mass water, for orthogonal and parallel polarisation, respectively). This phenomenon should be investigated more in depth, by carrying out further measurements on different kinds of wood (having different densities), in order to have a clear picture of it.

## 4 Conclusions

In this work, the sensitivity of Ground Penetrating Radar (GPR) signal to moisture variation in wood material was

investigated. The relative permittivity of an Epicea wood sample was measured, at different humidity levels and for different polarisation states of the electromagnetic field.

Results obtained by using direct waves in Wide Angle Radar Reflection (WARR) configuration, where one GPR antenna is moved while the other is in a fixed position, were compared to results obtained by using reflected waves, in the so-called Fixed Offset configuration where the distance between GPR antennas is fixed. Additionally, when the WARR

method was applied, it was investigated how the attenuation of the received signal varies as a function of the distance between the radar antennas.

The presented results prove that direct and reflected waves have different behaviours when the moisture content varies, due to their different propagation paths. Overall, when the humidity levels increase, the difference between the permittivity values estimated by using the reflected- and direct-wave approaches becomes larger.

For the reflected waves, the wood anisotropy affects the variation of the relative permittivity as a function of the moisture content; the effect is stronger when the electric field is parallel to the wood fibers. This is in good agreement with results available in the literature. Regarding direct waves, the measured values of the relative permittivity turn out to be weakly affected by the polarisation of the electromagnetic field. They are close to the values obtained by using reflected waves with electric field orthogonal to the wood fibers. Apparently, the propagation paths are similar in the two cases.

Overall, our results show that the proposed measurement approach is effective to estimate the permittivity behaviour of wood material as a function of moisture content. The GPR technique is then promising for moisture evaluation in timber structures and their early-stage diagnosis.

## Acknowledgment

The Authors are grateful to COST - European Cooperation in Science and Technology (www.cost.eu) for funding the Action

TU1208 *Civil engineering applications of Ground Penetrating Radar* (www.GPRadar.eu). The experimental results



presented in this paper were collected in the University of Bordeaux, France, during a 1-month Short-Term Scientific Mission supported by TU1208. The Authors are grateful to the EGU GI Division President, Dr Francesco Soldovieri, the Chief-Executive Editor of the *Geoscientific Instrumentation, Methods and Data Systems* journal, Dr Jothiram Vivekanandan, and the Executive Editors, Dr Ari-Matti Harri and Dr Håkan Svedhem, for inviting them to submit this paper, which resumes

and extends the presentation (Reci *et al*, 2016), given during the GI3.1 Session of the 2016 EGU GA (Vienna, Austria, 17-22 April 2016).

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

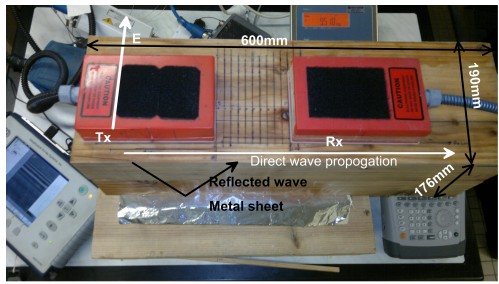

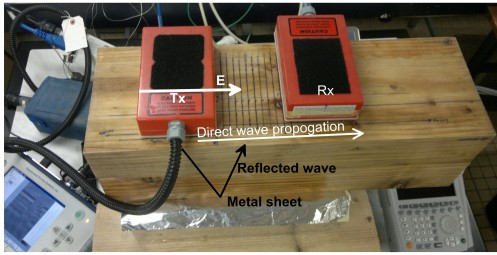

**Figure 1: GPR measurements on the wood sample, by using ground-coupled antennas and the direct-wave (WARR) technique. A)**
25 **Electric field is orthogonal to the wood fibers. B) Electric field is parallel to the wood fibers.**




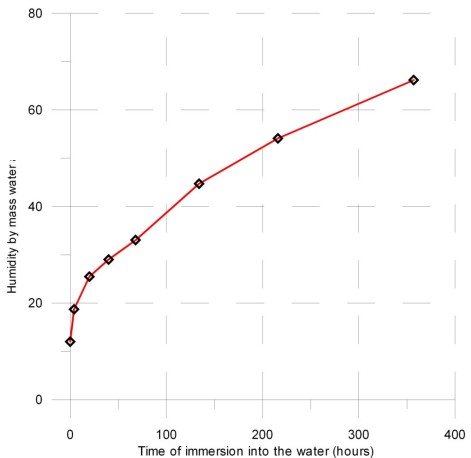

**Figure 2: Humidity by mass water as a function of the immersion time in water.**

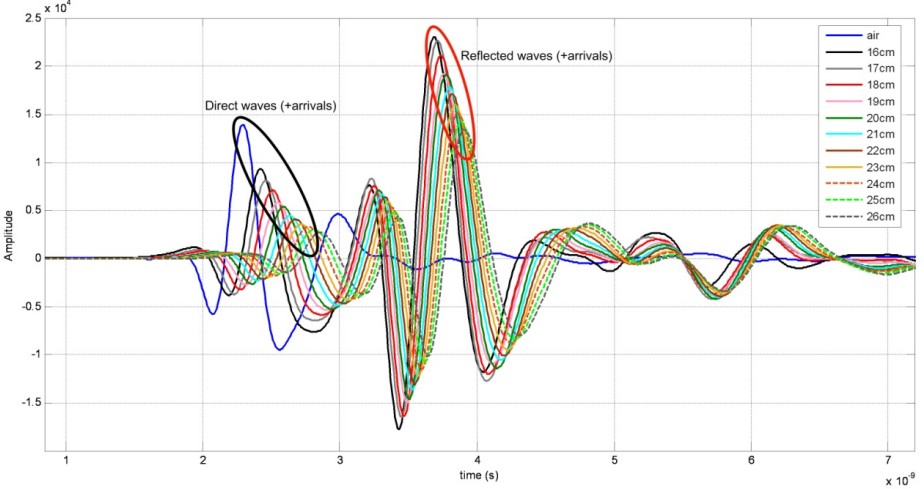

**Figure 3: Radargrams showing the superposition of the direct-air, direct and reflected waves measured over the sample, when the polarisation of the electric field is orthogonal to the fibers. In this case, the humidity level by mass water was 12%.**





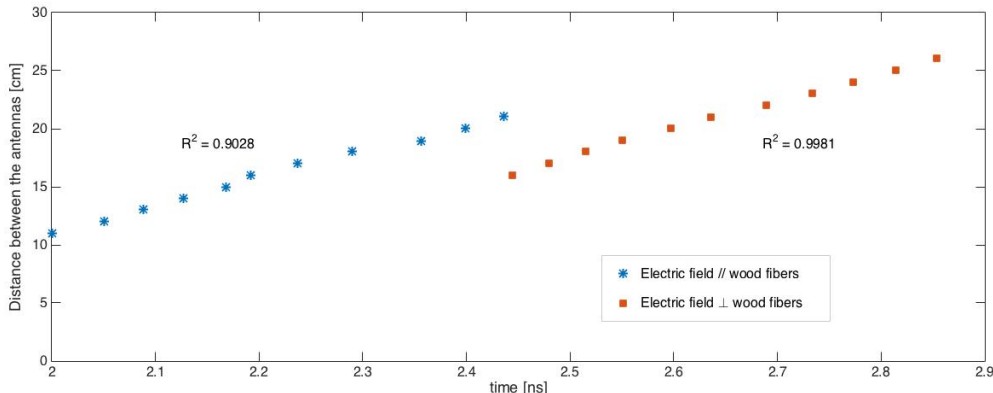

**Figure 4: Determination of the propagation velocity for the direct wave, from the arrival times. Both configurations with electric field parallel and orthogonal to the fibers are considered. In this case the humidity level was 18.18%.**

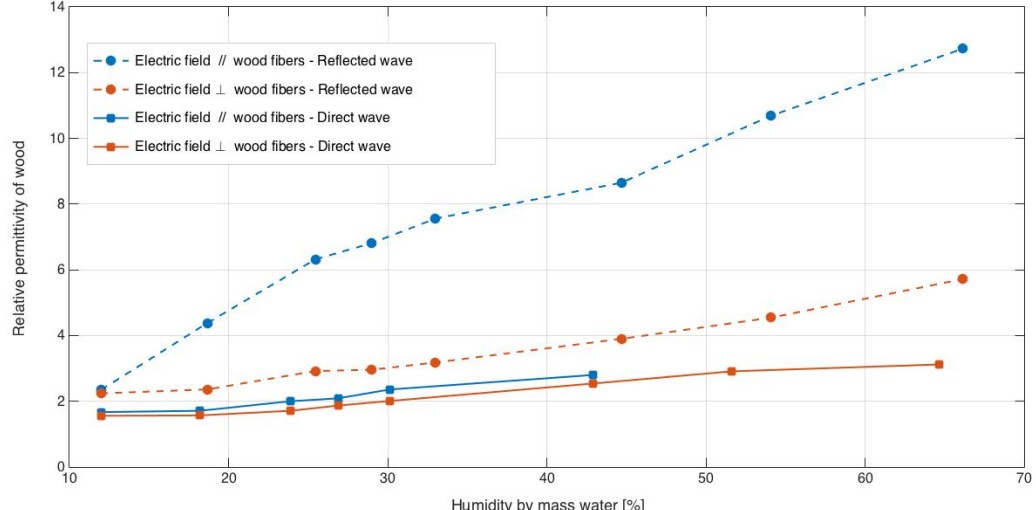

**Figure 5: Relative permittivity as a function of humidity by mass water, estimated by using the direct-wave (WARR) and reflected-wave methods, for both polarisation cases.**





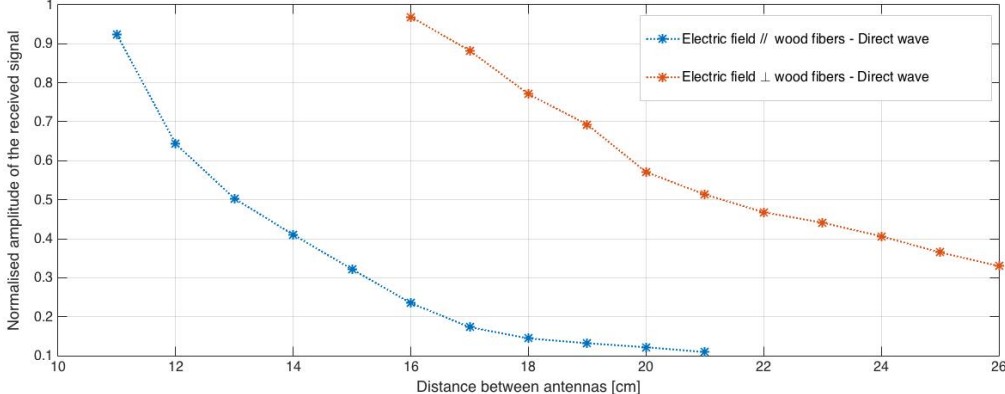

**Figure 6: Amplitude of the signal received by the GPR, as a function of the distance between transmitting and receiving antennas (WARR method, 18.18% humidity by mass water).**

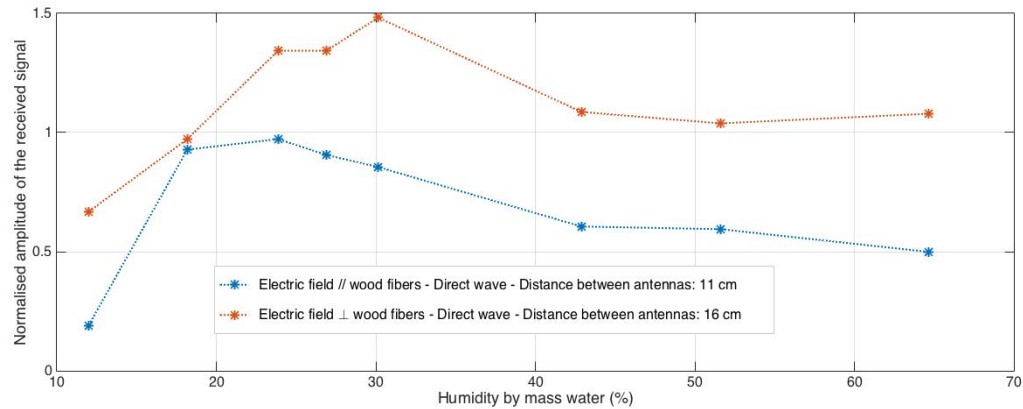

**Figure 7: Normalised amplitude of direct wave with respect to humidity, for orthogonal and parallel polarisation of E field.**



**Table I: Relative permittivity of wood, for different levels of humidity by mass water**

**and for direct- and reflected-wave methods.**

| Direct-wave (WARR) method | | | Reflected-wave method | | |
|---|---|---|---|---|---|
| Humidity (%) | E ⊥ wood fibers | E // wood fibers | Humidity (%) | E ⊥ wood fibers | E // wood fibers |
| 12 | 1.56 | 1.67 | 12 | 2.24 | 2.35 |
| 18.18 | 1.57 | 1.71 | 18.66 | 2.36 | 4.38 |
| 23.87 | 1.71 | 2 | 25.46 | 2.92 | 6.32 |
| 26.89 | 1.87 | 2.09 | 28.99 | 2.96 | 6.81 |
| 30.15 | 2.01 | 2.36 | 33.01 | 3.18 | 7.56 |
| 42.9 | 2.54 | 2.8 | 44.69 | 3.9 | 8.65 |
| 51.6 | 2.91 | - | 54.09 | 4.544 | 10.69 |
| 64.65 | 3.12 | - | 66.14 | 5.718 | 12.728 |