# Peer review of "Non-destructive evaluation of moisture content in wood by using Ground Penetrating Radar"

_Geoscientific Instrumentation, Methods and Data Systems, 2016_

## Referee Comment (RC1) · Anonymous Referee #1 · 26 Jul 2016

The paper is interesting, the authors might analyse somehow more in depth the reasons of the differences between the permittivity evacuate from the direct and from the reflected wave. In particular, they should say what measurement provides the best restults and why between the two. Moreover, the authors should say whether the moisture is uniform or it might exist some gradient of humidity in the wooden sampled: this might also influence the measure and in particular the difference between the results achieved from the direct wave and the reflected one.

---

## Short Comment (SC1) · 27 Jul 2016

The comment from referee 1, is very interesting. I am trying to answer to his comments shortly. The sampled wood is with almost uniform humidity, because the values of dielectric constants can be influenced if there is a gradient humidity too. Wood is an anisotropic media, so the dielectric properties of wood are strongly influenced form the polarization of Electric field in relation with wood grains, moreover these properties are influenced by cellulose and mannan in the longitudinal direction, but in transverse direction the dielectric properties are influenced by lignin. Lignin has lower dielectric properties than cellulose. Therefore, it is expected that transverse values are lower than longitudinal values for the same humidity, and this is more clear from the reflected wave. For the direct wave, the direction of propagation of the EM wave is almost as in the case of radial polarization so this change is lower. As a conclusion we can say

that the effect of wood species can be best studied under longitudinal direction and from the reflected wave, because the dielectric properties differentiate themselves as a function of the moisture content mos clearly in this direction. Whereas, this is not the case for the direct wave.
* * *

---

## Referee Comment (RC2) · Anonymous Referee #2 · 9 Sep 2016

This is an interesting experiment, which includes a brief introduction and a theoretical background of GPR followed by laboratory measurements and a brief summary. The whole paper gives the impression of a very quick and not very consistent work. From the beginning to the end there are many more or less serious inaccuracies. Considering those facts, the comments below, missing satisfactory explanations of the results, and a missing novelty of the paper, I regret not to recommend this paper for publishing.

Specifically: English should be improved (ideally checked by a native speaker).

GPR image of some measurement would be welcome. In this paper, we can see only A-scans.

In Table I there are no symbols ($\varepsilon$r) and units ([-]) for the relative permittivity.

In Figure 4 we can see a variable R (R2=0.9028, R2=0.9981) without any description of its meaning. Is it supposed to be a measure of uncertainty/goodness of fit (sometimes also denoted as ˆ2)?

Captions and axis labels in figures should be of the same size and should start with a capital letter.

Brackets in all axis labels should be unified.

The captions in figures should be unified with the text, sometimes the important information is only in the captions and sometimes only in the text.

Not many measurements were done and so the data cannot be considered as decisive.

Page1: Line 26: In this paper, there is a sentence "For example, it is known that the minimum value of moisture content for the development of wood degrading fungi is 17% by mass, whereas the optimum values range from 30% to 70% depending on the fungi and wood type (Mai et al., 2015)." which was basically copied from the cited article of Mai et al., 2015. Unfortunately, this statement is not proved by any reference in that paper. Could the authors back up those statements? Myself, I found different values of moisture which are optimal for fungi development. The formulation "it is known" is then not very well chosen in this case. The authors claim, that the moisture content is the most common cause of wood deterioration. Do they know other factors influencing biological degradation of wood caused by fungi and insects? Out of curiosity, do the authors know, how can be the moisture of wood changed by the fungi attack?

Page 3: Line 1: Speed of light in vacuum is approximately equal to 0.3 m/ns (more precisely 299 792 458 m/s), it is not equal to 0.3 m/ns and it is also not speed of light in the air (On Earth, Mars, or Titan?).

Page 3: Line 3: There is a mismatch in terminology. Symbol $\varepsilon r$ is the relative permittivity (called also as the dielectric constant), it is confusing to call it "dielectric relative permittivity" or "dielectric permittivity" in the line 13 of the same page, etc.

Page 3: Line 6: There is, again, a mismatch in terminology. Symbol $\varepsilon$ is the permittivity or absolute permittivity, it is not "the dielectric permittivity" claimed by authors.

Page 3: Line 6: The vacuum permittivity $\varepsilon_0$ (or electric constant, or permittivity of free space) is not equal but is approximately equal to 8.854187*10-12 F/m. Nevertheless, I assume that the "$\varepsilon_0$=8.854*1012 F/m" is just a typo.

Page 3: Line 7: The authors claim, that wood is a low-lossy medium. Is it true for all humidity levels of wood? Is it also true for living trees? In what range do the authors define low-lossy media and what are high-lossy media according to them?

Page 3: Line 16: The Equation (2) does not seem to be correct. Furthermore, it is missing in the reference given by authors, Neal, 2004. Could authors provide more information about the equation derivation?

Page 3: Line 27: The authors chose the initial humidity level equal to 12%. Why did they choose this value of humidity? How did they measure that?

Page 3: Line 30: The Equation (3) given by Moron et al., 2016 is strictly determined for samples in the anhydrous state for W0. In spite of that, authors used the sample with the humidity level of 12%, which is not correct and could (and probably did) highly influence their further data.

Page 4: Line 2: "The weight of the sample is measured with a balance having a sensitivity to the gram." It is a useless information if we do not know the mass of the measured sample in the anhydrous state. For example, if the sample has 3 grams, such a measurement will be entirely insufficient.

Page 4. Line 4: "...an almost curvilinear increase occurs." The authors do not have enough data points to say anything about the shape of the curve. It appears to me more like linear.

Page 4. Line 20: In the Equation (4), v0 is not defined.

Page 4. Line26: It would be much clearer if the authors provided a simple drawing of the disposition of the measurement. It is not immediately obvious what dR, dTR, and h mean.

Page 4. Line29: If the Equations (4) and (5) are exact, then the Equation (7) is exact as well. There is equality, not estimation.

Page 4. Line29: Why the velocity v in Equation (7) has a different style than previously (italic, non-italic)?

Page 5. Line1: There is a note about the humidity level of the sample "ranging from 12% to 64.65%" which is incomprehensibly precise. The humidity levels are calculated from a hydrated sample and then, the humidity is basically estimated and precision like that is not required.

Page 5. Line6: Could authors explain why only the reflected wave with the parallel polarization shows such a different increasing of the relative permittivity than the others?

Page 5. Line 8: "The increase of relative permittivity versus moisture content is piecewise linear, with a slope change occurring when the humidity level is about 18%." This statement is based on single measurement when the humidity was around 18%. There is no measurement error on the point, the measurement at 18% can be just a small upward fluctuation.

Page 5. Line 8 and 9: "Moisture content is piecewise linear." Of course, it is piecewise linear, if the graph is made by connecting data points from isolated measurements. Why did the authors not make a fit instead of connecting points?

Page 5. Line 19: "Apparently, the propagation paths are similar in the two cases." Do the authors have some explanation, why is it like that? Did they try to adapt their equations for two cases?

Page 5. Line 25: The authors claim that the last possible humidity level which can be measured is 43%. Did the authors try the measurement with the humidity of 44% or

45%? Or did they just tested 43% as the last point where it worked, and then 52% and it did not work?

Page 5. Line 32: Why did the authors mix the distances of the measurement in Figure 7? We cannot really compare two different measurements when the distance (and so the attenuation) is greater.

Page 6. Line 3: "...this may be due to a superposition of direct and direct air waves." Did the authors try other calculations or simulation to fully describe the phenomenon? Why does it not occur in the other case? Do they have any explanation why does the amplitude firstly increase and then decrease?

Page 6. Line 25: The authors claim that the measurement is effective to estimate the permittivity behavior regardless of different results by two different techniques (especially the reflective wave method). Do they have some error tolerance for the permittivity determination?

Please also note the supplement to this comment:
http://www.geosci-instrum-method-data-syst-discuss.net/gi-2016-24/gi-2016-24-RC2-supplement.pdf

---

## Short Comment (SC2) · 14 Sep 2016

I have read in carefully, the comments of Referee #2 related to the manuscript "Non-destructive evaluation of moisture content in wood by using Ground Penetrating Radar" H. Reci, T. C. Maï, Z. M. Sbartaï, L. Pajewski, and E. Kiri We thank a lot the referee for the detailed suggestions about the manuscript and we agree that improvement should be done. Of course the manuscript needs corrections and we are working to improve it and make the necessary correction in the new version of the manuscript. I have those comments: We can include B scans from the measurements of course. We decided to exclude them in order to not put so many figures in the text. However attached you will see images from measurements and B scans for different levels of Humidity by mass water and polarization of E field (figure 1). Corrections regarding the axes, symbols and so on.. will be fixed as suggested by the referee. As related to the humidity

12%, and how did we determine that, we can say that we found at the end of the experiments putting the sample to the oven and finding the weight in anhydrous mode. Some corrections need to be done on mistyping and formulas, such as R2, which is the correlation coefficient of the measurements, or what each variable represent on the formulas. Of course there are other factors influencing biological degradation of wood caused by fungi and insects except moisture, like temperature, relative humidity and so on which is beyond the target of this work,,, but the scope of the paper was to be focused on influence of the moisture content on the dielectric properties of wood and the difference between reflected and direct wave. The comments by the referee will be seen and corrected in the new version of the manuscript most of them are orthographic and some of them mistyped. As a conclusion i would like to say that another referee comment is needed after working on article and uploading the new version.
* * *
[Figure]

A) Direct air, direct wave and reflected wave, for 12% humidity. Electric field is polarized perpendicular to fibers

[Figure]

B) Direct air, direct sample and reflected wave for 22 % humidity. Electric field is polarized perpendicular to the fibers.

[Figure]

C) Direct air, direct sample and reflected wave for 12% humidity. Electric field is polarized parallel to fibers

**Fig. 1.** B scans

---

## Referee Comment (RC3) · Anonymous Referee #1 · 16 Sep 2016

I have read the replaies to botyh referees. In my opinion, the updated paper can be now published as it is.

---

## Short Comment (SC3) · 16 Sep 2016

We really appreciate the comment by the referee #3. Uploaded paper will have all the necessary correction and some mistyped symbols and characters.

---

## Author Comment (AC1) · 24 Oct 2016

**Final comment on *Non-destructive evaluation of moisture content in Wood by using Ground Penetrating Radar*, by Hamza Reci et al.**

**1. Anonymous Referee #1:**

**Comment of Anonymous Referee #1:**
The paper is interesting, the authors might analyze somehow more in depth the reasons of the differences between the permittivity evacuate from the direct and from the reflected wave. In particular, they should say what measurement provides the best results and why between the two. Moreover, the authors should say whether the moisture is uniform or it might exist some gradient of humidity in the wooden sampled: this might also influence the measure and in particular the difference between the results achieved from the direct wave and the reflected one.

**Answers of authors to Anonymous Referee # 1:**
The comment of Referee 1 is very interesting. The sampled wood is with almost uniform humidity; of course the permittivity would be influenced by a gradient of humidity and we wanted to avoid this. Wood is an anisotropic medium, therefore it shows different dielectric properties if the polarisation of the electric field changes. The dielectric properties of wood are influenced by cellulose and mannan in the longitudinal direction, by lignin in the transverse direction. Lignin has lower dielectric properties than cellulose. Therefore, it is expected that transverse values are lower than longitudinal values for a fixed humidity level. This is more evident by looking at the reflected wave. For the direct wave, the direction of propagation of the electromagnetic wave is almost the same as in the case of radial polarization so a lower difference is observed. As a conclusion we can say that the effect of wood species can be best studied under longitudinal direction and by looking at the reflected wave, because the dielectric properties differentiate themselves as a function of the moisture content most clearly in this direction. In the case of the direct wave this change is smaller.

**2. Anonymous Referee #2**

**Comment of Anonymous Referee #2:**
This is an interesting experiment, which includes a brief introduction and a theoretical background of GPR followed by laboratory measurements and a brief summary. The whole paper gives the impression of a very quick and not very consistent work. From the beginning to the end there are many more or less serious inaccuracies. Considering those facts, the comments below, missing satisfactory explanations of the results, and a missing novelty of the paper, I regret not to recommend this paper for publishing.
Specifically: English should be improved (ideally checked by a native speaker).
GPR image of some measurement would be welcome. In this paper, we can see only A-scans.

**Answer of authors:**
Thank you for these comments. This is a short paper having the objective of analyzing the potential of the direct wave to evaluate moisture in wood material. It is important to note that there are no papers, in the literature, on this topic. We think that the presented results are novel and worth to be published, although they represent a first analysis and the issue needs to be studied more in depth. The paper was revised carefully and our results were compared with results available in the literature and concerned with the use of reflected wave. The English was improved and GPR images were added.

**Comment of Anonymous Referee #2:**
In Table I there are no symbols (Epsr) and units ([-]) for the relative permittivity.

**Answer of authors:**
There is no unit for the relative permittivity; the symbol 'r' was added.

**Comment of Anonymous Referee #2:**
In Figure 4 we can see a variable R (R2=0.9028, R2=0.9981) without any description of its meaning.
Is it supposed to be a measure of uncertainty/goodness of fit (sometimes also denoted as ˆ2)?
Captions and axis labels in figures should be of the same size and should start with a capital letter.

**Answer of authors:**
$R^2$ is the correlation coefficient of the measurements.
$R^2$ was corrected to be 0,90 and 0,99.
The captions and axis labels were corrected.

**Comment of Anonymous Referee #2:**
Brackets in all axis labels should be unified.

**Answer of authors:**
Done.

**Comment of Anonymous Referee #2:**
The captions in figures should be unified with the text, sometimes the important information is only in the captions and sometimes only in the text.

**Answer of authors:**
Done.

**Comment of Anonymous Referee #2:**
Not many measurements were done and so the data cannot be considered as decisive.

**Answer of authors:**
These are our first results and the work will continue. We believe they are worth to be published because no papers exist on the use of the direct wave for this application. Moreover, in the revised manuscript the dielectric constant obtained with the reflected wave is now compared with the literature, with good agreement. We believe this enriches the paper.

**Comment of Anonymous Referee #2:**
Page1: Line 26: In this paper, there is a sentence "For example, it is known that the minimum value of moisture content for the development of wood degrading fungi is 17% by mass, whereas the optimum values range from 30% to 70% depending on the fungi and wood type (Mai et al., 2015)." which was basically copied from the cited article of Mai et al., 2015. Unfortunately, this statement is not proved by any reference in that paper. Could the authors back up those statements? Myself, I found different values of moisture which are optimal for fungi development. The formulation "it is known" is then not very well chosen in this case. The authors claim, that the moisture content is the most common cause of wood deterioration.

Do they know other factors influencing biological degradation of wood caused by fungi and insects? Out of curiosity, do the authors know how can be the moisture of wood changed by the fungi attack?

**Answer of authors:**
Of course there are other factors affecting the biological degradation of wood caused by fungi and insects, moisture is not the only one. For example, some factors are the source of infection, substrate (food), oxygen, temperature, etc, but these are beyond the target of this work. The scope of the paper was to study the influence of moisture content on the dielectric properties of wood, and compare results obtained by using the reflected and direct waves.
Once started, fungi can produce a certain amount of moisture by the chemical decomposition of the wood, thus increasing the moisture content of wood if the evaporation loss is low.
Fungal spores do not germinate readily on wood if the moisture content is below the fiber saturation point; the latter is commonly reached around 25 to 30%. The percentage of moisture that is required for wood rotting fungi to flourish, depends on the species of fungi and the kind of wood. In the revised version the relevant statements were corrected.

**Comment of Anonymous Referee #2:**
Page 3: Line 1: Speed of light in vacuum is approximately equal to 0.3 m/ns (more precisely 299 792 458 m/s), it is not equal to 0.3 m/ns and it is also not speed of light in the air (On Earth, Mars, or Titan?).

**Answer of authors:**
We wrote 30 cm/ns, which is 0.3 m/ns. As is very well known, this is the speed of light in a vacuum and it is approximately equal to the speed of light in the air. We agree that is better to be more precise, therefore we changed speed in the air with speed in a vacuum.

**Comment of Anonymous Referee #2:**
Page 3: Line 3: There is a mismatch in terminology. Symbol "r is the relative permittivity(called also as the dielectric constant), it is confusing to call it "dielectric relative permittivity" or "dielectric permittivity" in the line 13 of the same page, etc.

**Answer of authors:**
We agree, the relative permittivity is obviously a complex number and its real part is the dielectric constant. Dielectric relative permittivity was changed with "dielectric constant" in the revised manuscript.

**Comment of Anonymous Referee #2:**
Page 3: Line 6: There is, again, a mismatch in terminology. Symbol " is the permittivity or absolute permittivity, it is not "the dielectric permittivity" claimed by authors.

**Answer of authors:**
This is the dielectric constant. The correction was done in the revised manuscript.

**Comment of Anonymous Referee #2:**
Page 3: Line 6: The vacuum permittivity "0 (or electric constant, or permittivity of free space) is not equal but is approximately equal to 8.854187*10-12 F/m. Nevertheless, I assume that the ""eps0=8.854*1012 F/m" is just a typo.

**Answer of authors:**
This was a typo, eps0=8.854*10⁻¹² F/m . The correction was done in the revised manuscript.

**Comment of Anonymous Referee #2:**
Page 3: Line 7: The authors claim, that wood is a low-lossy medium. Is it true for all humidity levels of wood? Is it also true for living trees? In what range do the authors define low-lossy media and what are high-lossy media according to them?

**Answer of authors:**
A low-lossy medium has eps"/eps' << to 1 (as concrete) and there is not a specific limit value. We used this assumption to simplify the mathematical formula that relates the velocity to the dielectric constant.

**Comment of Anonymous Referee #2:**
Page 3: Line 16: The Equation (2) does not seem to be correct. Furthermore, it is missing in the reference given by authors, Neal, 2004. Could authors provide more information about the equation derivation?

**Answer of authors:**
We are surprised by this comment because this formula is well known and used by researchers for calculating the dielectric constant by measuring the velocity of electromagnetic wave as GPR. We present in the following equations the derivative and details calculation of the velocity:

$$v = v_p = \left(\frac{dz}{dt}\right)_{\varphi=const} = \frac{\omega}{\beta}$$

The phase coefficient:

$$\beta = \omega\sqrt{\frac{\mu\left(\sqrt{\varepsilon_e'^2 + \varepsilon_e''^2} + \varepsilon_e'\right)}{2}} = \omega\sqrt{\frac{\mu\varepsilon_0\varepsilon_r'}{2}\left(\sqrt{1 + \frac{\varepsilon_r''^2}{\varepsilon_r'^2}} + 1\right)} = \frac{\omega}{c}\sqrt{\frac{\mu_r\varepsilon_r'}{2}\left(\sqrt{1 + \frac{\varepsilon_r''^2}{\varepsilon_r'^2}} + 1\right)}$$

Then :

$$v = v_\varphi = \frac{\omega}{\beta} = \frac{c\sqrt{2}}{\sqrt{\mu_r\left(\sqrt{\varepsilon_r'^2 + \varepsilon_r''^2} + \varepsilon_r'\right)}}$$

If we assume that $\frac{\varepsilon_r''}{\varepsilon_r'} \ll 1$ then:

$$\beta \approx \omega\sqrt{\mu_0\varepsilon_e'} \qquad \beta \approx \omega\sqrt{\mu_0\varepsilon_0\varepsilon_r'} the \quad \beta \approx \frac{\omega}{c}\sqrt{\varepsilon_r'}$$

$$v = v_\varphi = \frac{\omega}{\beta} \approx \frac{1}{\sqrt{\mu_0\varepsilon_0\varepsilon_r'}} then v = v_\varphi \approx \frac{c}{\sqrt{\varepsilon_r'}}$$

These formulas were added to the revised manuscript.

**Comment of Anonymous Referee #2:**
Page 3: Line 27: The authors chose the initial humidity level equal to 12%. Why did they choose this value of humidity? How did they measure that?

**Answer of authors:**
This humidity level is known to be the reference humidity level (water content) used for wood characterization, that is why we used this humidity level. We measured that the reference humidity was 12%, at the end of the experiments putting the sample to the oven and finding the weight in anhydrous mode.

**Comment of Anonymous Referee #2:**
Page 3: Line 30: The Equation (3) given by Moron et al., 2016 is strictly determined for samples in the anhydrous state for W0. In spite of that, authors used the sample with the humidity level of 12%, which is not correct and could (and probably did) highly influence their further data.

**Answer of authors:**
Yes is true but the influence is not very significant. To take into account this effect, the calculated density at zero humidity is used to correct the humidity levels.

**Comment of Anonymous Referee #2:**
Page 4: Line 2: "The weight of the sample is measured with a balance having a sensitivity to the gram." It is a useless information if we do not know the mass of the measured sample in the anhydrous state. For example, if the sample has 3 grams, such a measurement will be entirely insufficient.

**Answer of authors:**
Yes but the sample is of 60x20x20 cm and the density of the tested wood is about 450 kg/m$^3$. Then, the sample at zero humidity has a mass of about 10 kg. The error due to balance precision is of about 0,01%. This information was added in the revised manuscript.

**Comment of Anonymous Referee #2:**
Page 4. Line 4: ": : :an almost curvilinear increase occurs." The authors do not have enough data points to say anything about the shape of the curve. It appears to me more like linear.

**Answer of authors:**
Yes you are right but we know by experience (see paper Maï and all, CBM, 2012) that the shape is curvilinear. As you suggested, in the revised manuscript we removed this sentence.

**Comment of Anonymous Referee #2:**
Page 4. Line 20: In the Equation (4), v0 is not defined.

**Answer of authors:**
It is defined before; it is the velocity of light in a vacuum.

**Comment of Anonymous Referee #2:**
Page 4. Line26: It would be much clearer if the authors provided a simple drawing of the disposition of the measurement. It is not immediately obvious what dR, dTR, and h mean.

**Answer of authors:**
Done.

**Comment of Anonymous Referee #2:**
Page 4. Line29: If the Equations (4) and (5) are exact, then the Equation (7) is exactas well. There is equality, not estimation.

**Answer of authors:**
Done.

**Comment of Anonymous Referee #2:**
Page 4. Line29: Why the velocity v in Equation (7) has a different style than previously (italic, non-italic)?

**Answer of authors:**
Ok, corrected.

**Comment of Anonymous Referee #2:**
Page 5. Line1: There is a note about the humidity level of the sample "ranging from12% to 64.65%" which is incomprehensibly precise. The humidity levels are calculated from a hydrated sample and then, the humidity is basically estimated and precision like that is not required.

**Answer of authors:**
Ok it was changed to "ranging from 12% to 64,5%."

**Comment of Anonymous Referee #2:**
Page 5. Line6: Could authors explain why only the reflected wave with the parallel polarization shows such a different increasing of the relative permittivity than the others?

**Answer of authors:**
Wood is an anisotropic media, so its dielectric properties are strongly influenced by the polarization of the electric field in relation with wood grains, moreover these properties are influenced by cellulose and mannan in the case of parallel polarization, whereas in transverse direction the dielectric properties are influenced by lignin. Lignin has lower dielectric properties than cellulose. Therefore, it is expected that the values of the dielectric constant in parallel polarization are more influenced from the humidity than in transverse direction. In the case of the reflected wave, the electric field can be polarized exactly as the wood grains.

**Comment of Anonymous Referee #2:**
Page 5. Line 8: "The increase of relative permittivity versus moisture content is piecewise linear, with a slope change occurring when the humidity level is about 18%." This statement is based on single measurement when the humidity was around 18%. There is no measurement error on the point, the measurement at 18% can be just a small upward fluctuation.

**Answer of authors:**
This is in agreement with a previous publication where more samples were considered and the results showed the same behavior with a slope change around 18 to 20% of moisture.

**Comment of Anonymous Referee #2:**
Page 5. Line 8 and 9: "Moisture content is piecewise linear." Of course, it is piecewise linear, if the graph is made by connecting data points from isolated measurements. Why did the authors not make a fit instead of connecting points?

**Answer of authors:**
It is not necessary because the objective is not to find the fit between dielectric constant and humidity but to compare direct and reflected waves results.

**Comment of Anonymous Referee #2:**
Page 5. Line 19: "Apparently, the propagation paths are similar in the two cases."
Do the authors have some explanation, why is it like that? Did they try to adapt their equations for two cases?

**Answer of authors:**
When the electric field is polarized in transverse direction the dielectric properties of wood are influenced by lignin. Lignin has lower dielectric properties and this could be the reason for this small change.

**Comment of Anonymous Referee #2:**
Page 5. Line 25: The authors claim that the last possible humidity level which can be measured is 43%. Did the authors try the measurement with the humidity of 44% or45%? Or did they just tested 43% as the last point where it worked, and then 52% and it did not work?

**Answer of authors:**
No, we tested more values but the signal is completely attenuated in the parallel polarization and then it is not possible to extract signal parameters.

**Comment of Anonymous Referee #2:**
Page 5. Line 32: Why did the authors mix the distances of the measurement in Figure7? We cannot really compare two different measurements when the distance (and so the attenuation) is greater.

**Answer of authors:**
We can't use the same distance because with parallel polarization the attenuation is very high compared to perpendicular polarization. However, the attenuation curves can be compared because whatever the distance the calculation of the attenuation is similar.

**Comment of Anonymous Referee #2:**
Page 6. Line 3: ": : :this may be due to a superposition of direct and direct air waves."Did the authors try other calculations or simulation to fully describe the phenomenon? Why does it not occur in the other case? Do they have any explanation why does the amplitude firstly increase and then decrease?

**Answer of authors:**

It occurs only in this case because the distance T-R is too small and the dielectric constant is also small, therefore the velocity is high and the time of flight is too small in this configuration that allows to a superposition.

**Comment of Anonymous Referee #2:**

Page 6. Line 25: The authors claim that the measurement is effective to estimate the permittivity behavior regardless of different results by two different techniques (especially the reflective wave method). Do they have some error tolerance for the permittivity determination?

**Answer of authors:**

The tolerance is presented in another paper, such reference was added regarding the evaluation of dielectric constant by reflected wave (Chinh et al, 2015). However, in our case the objective is to show if the direct wave is able to evaluation the humidity, and the answer is that the direct wave is less affected by moisture than the reflected wave because of the anisotropy of wood.

**3. Anonymous Referee #1**

**Comment of Anonymous Referee #1**

I have read the replays to both referees. In my opinion, the updated paper can be now published as it is.

**Answers to referee #1**

We really appreciate the comment by the referee #1. The revised manuscript includes all the necessary corrections and integrations, as described above.

[revised manuscript text omitted]

Consider a plane electromagnetic wave propagating through wood in the z-direction. If losses due to conductivity and reflections are ignored, the attenuation and wavelength are governed by the factor $e^{-\gamma z}$, with complex propagation constant,

$$\gamma = j\omega\sqrt{\varepsilon^* \varepsilon_0 \mu^{'} \mu_0} = a + j\beta \tag{1}$$

where $\omega$ is the angular frequency of the wave, $\varepsilon^* = \varepsilon^{'} - j\varepsilon^{''}$ is the complex dielectric costant, $\mu^{'} \approx 1$ is the relative permeability of wood, and $\varepsilon_0 = 1/(\mu_0 c_0^2)$ is the permittivity of free space where $c_0$ is the speed of the light in free space. The real part of $\gamma$ can be defined as an attenuation constant,

$$\alpha = \frac{\omega}{c_0}\left[\frac{\varepsilon^{'}}{2}\left(\sqrt{1 + tan^2\delta} - 1\right)\right]^{\frac{1}{2}} \tag{2}$$

and the imaginary part of $\gamma$ as a phase constant,

$$\beta = \frac{\omega}{c_0}\left[\frac{\varepsilon^{'}}{2}\left(\sqrt{1 + tan^2\delta} + 1\right)\right]^{\frac{1}{2}} \tag{3}$$

where $tan\delta = \varepsilon^{'}/\varepsilon^{''}$ is defined as loss tangent.

Electromagnetic waves propagate in the free space at an approximate speed of 0.3 m/ns. In wood, which is a dielectric anisotropic material, the electromagnetic velocity can be expressed as follows:

$$v = \frac{\omega}{\beta} \approx \frac{c}{\sqrt{\varepsilon_r}} \tag{4}$$

where $\varepsilon_r = \varepsilon/\varepsilon_0$ is the relative permittivity, $\varepsilon_0 = 8.854 \cdot 10^{-12}$ F/m is the permittivity in a vacuum.

In this work, GPR radargrams are recorded by using two different techniques: direct and reflected waves are measured by using the Wide Angle Radar Reflection (WARR) and Fixed Offset (FO) methods, respectively, as will be explained in the following. Ground-coupled antennas are employed, with central frequency 1.5 GHz; the radar system is a GSSI SIR 3000. A wood sample of Epicea (Spruce) type is used, which is 600 mm long, 190 mm wide and 176 mm thick (see Figure 1). Measurements are carried out in two directions: longitudinal (as in Figure 1.A), where the electric field is 25 polarized orthogonal to the wood fibers, and transversal (as in Figure 1.B), where the electric field is parallel to the fibers. In order to easily distinguish between reflected and direct waves, a metallic sheet is placed under the wood sample.

Measurements start at a humidity level of the wood sample equal to 12%, known as the reference humidity (water content) used for wood characterization. We measured that the reference humidity was 12%, at the end of the experiments putting the sample to the oven and finding the weight in anhydrous mode. The calculated density at zero humidity is used to correct the humidity values. Afterwards, the sample is immersed into the water in order to gradually increase its moisture content. GPR experiments are then repeated at different humidity levels.

Humidity by mass water (%) is calculated by adopting the following expression (Moron $et\ al.$, 2016):

$$Humidity\ (\%) = \left(\frac{W - W_0}{W_0}\right) 100 \qquad (5)$$

where $W_0$, is the weight of the sample in anhydrous mode, W is the weight of the sample after being immersed into the water. The weight of the sample was measured with a balance having sensitivity to the gram. The sample is of 60x19x17.6 cm and the density of the tested wood is about 450 kg/m³ so the the sample at zero humidity has a mass of about 10 kg and the error due to balance precision is of about 0,01%.

Figure 2 shows the humidity by mass water of the sample, as a function of the time of immersion into the water.

The measurements are performed at the humidity levels listed in Table I.

When applying the WARR technique, a radar antenna is kept in a fixed position and the other antenna is moved on the wood surface with a 1 cm step. The distance between the two antennas varies from 16 to 26 cm and from 11 to 21 cm, for orthogonal and parallel polarization of the electric field, respectively. When the FO method is applied, the distance between the antennas is 16 and 11 cm for orthogonal and parallel polarization states, respectively. The arrival times are visualized with Radan Software and Matlab. Examples of A-scans and B-scans showing the superposition of direct air wave, direct wave and reflected wave are provided in Figure 3&4; here, the polarization of the electric field is orthogonal to the fibers and the humidity level by mass water is 12%.

For the WARR technique, the propagation velocity is estimated from the arrival times of the direct waves, measured at difference distances between the antennas (the arrival time is the instant corresponding to the first and highest positive peak in the radargram). In particular, the propagation velocity is estimated as the slope of the linear regression of the arrival time of the direct wave, as a function of the distance between antennas. This is shown in Figure 5, for both polarization cases and a level of humidity equal to 18.18%.

For the reflected wave, the propagation velocity $v$ in the wood sample was determined from the peaks of the air wave (+D) and reflected wave (+R) and the following expressions were used (Figure 4). For the direct air wave (+D), the arrival time is:

$$t_{air} = t_0 + t_{air}^{real} = t_0 + \frac{d_{TR}}{v_0} \qquad (6)$$

where $t_{air}^{real}$, is the arrival time of the air wave (reference signal), $t_0$ is the starting time of electromagnetic impulse, $v_0$ is the velocity of the light in the air, and $d_{TR}$, is the distance between the transmitting and receiving antennas.

For the reflected wave (+R) the arrival time is:

$$t_r = t_0 + t_r^{real} = t_0 + \frac{d_R}{v} \qquad (7)$$

where $t_r^{real}$, is the arrival time of the reflected wave and $d_R$ is the length of the propagation path of the reflected wave, which is given by the equation:

$$d_R = 2\sqrt{(\frac{d_{TR}}{2})^2 + h^2} \qquad (8)$$

where h, is the thickness of the wood sample (Figure 6).

From the combination of Equations (7) and (8) it is possible to find the propagation velocity inside the wood sample, as follows:

$$v = \frac{d_R}{\Delta t + \frac{d_{TR}}{v_0}} \qquad (9)$$

where $\Delta t = t_r^{real} - t_{air}^{real}$. Finally, the relative permittivity of the wood sample can be estimated from the following expression:

$$\varepsilon' \approx (v_0/v)^2 \qquad (10)$$

**3 Results and discussions**

As mentioned in Section 2, the wood relative permittivity is measured for different humidity levels (ranging from 12% to 64.5%) and polarization cases (electric field orthogonal and parallel to the wood fibers). Results are summarized in Table I and plotted in Figure 7.

When the direct-wave method is used, the estimated value of the relative permittivity does not significantly change if the polarization is rotated. When the electric field is parallel to the fibers, the permittivity values are systematically higher than those measured when the electric field is orthogonal to the fibers. Wood is an anisotropic media, so the dielectric properties of it are strongly influenced from the polarization of Electric field in relation with wood grains, moreover these properties are influenced by cellulose and mannan in the case of parallel polarization, but in transverse direction the dielectric properties are influenced by lignin. Lignin has lower dielectric properties than cellulose. Therefore, it is expected that the values of dielectric constants from the parallel polarization are more influenced from the humidity than in transverse direction. In the case of the reflected wave the Electric field can be polarized exactly to the wood grains.

The increase of relative permittivity versus moisture content is piecewise linear, with a slope change occurring when the humidity level is about 18%, which is in agreement with with previous publication with more samples and the results

For the reflected-wave method, the increase of relative permittivity versus moisture content is piecewise linear as well, with a higher slope than in the case of the direct-wave method. Moreover, the slope does strongly depend on the polarization of the electromagnetic field and this is in agreement with (Martinez *et al.*, 2013b, and Mai *et al.*, 2015). When the electric field is orthogonal to the wood fibers, a slope change occurs at a humidity level of about 18%, corresponding to the fiber saturation point. The slope change is less visible and seem to occur at higher humidity levels, when the electric field is parallel to the wood fibers; this is again in agreement with (Martinez *et al.*, 2013b, and Mai *et al.*, 2015).

At all humidity levels, the permittivity values measured by the reflected-wave method are consistently higher than those measured by the direct-wave method. For both methods, the direction of the fibers does not affect the wood permittivity when the moisture content is low, and then it becomes more important in the presence of higher humidity levels.

It is interesting to notice that the results of the reflected-wave method are closer to the direct-wave curves when the electric field is orthogonal to the wood fibers. When the electric field is polarized in transverse direction the dielectric properties of wood are influenced by lignin. Lignin has lower dielectric properties and this could be the reason for this small change.

The obtained results show that direct waves in wood behave differently than reflected waves. This happens because the direct and reflected waves follow different propagation paths: the direct waves propagate in the top layer of the sample and the effect of the electromagnetic-field polarization is small; the reflected waves propagate through the whole sample and, due to the anisotropy of wood material, the polarisation has a stronger effect on the results.

When the electric field is orthogonal to the wood fibers, direct waves can be distinguished even when the humidity levels are above 60%. When the electric field is parallel to the wood fibers, instead, the direct wave arrival time cannot be detected for humidity levels higher than 43%. We tested more for humidity levels higher than 43%, but the signal is completely

attenuated in the parallel polarization and then it is not possible to extract signal parameters. Indeed, a high dissipation of electromagnetic energy occurs and the waves are highly attenuated.

A further goal of this work is to study how the distance between the radar antennas affects the amplitude of the received signal. For each considered humidity level, the amplitude of the direct wave is then measured with antennas placed 30 at
5  different distances. In Figure 8, the direct-wave amplitude normalized to the amplitude of the direct air wave is plotted, as a function of the distance between transmitting and receiving antennas, when the humidity by mass water is 18.18%. As expected, the amplitude shows an exponential attenuation when the distance increases. In Figure 9, the normalized amplitude of the direct wave is plotted as a function of the humidity level, for both parallel and orthogonal polarization cases, when the distance between the antennas is 11 cm and 16 cm, respectively. It can be noticed that, when the moisture content increases,
10  the normalized amplitude at small distances turns out to be higher than one, when the electric field is orthogonal to the wood fibers: this may be due to a superposition of direct and direct air waves. It occurs because the distance T-R is too small and the dielectric constant is also small then the velocity is high and the time of flight is too small in this configuration that allows to a superposition.

[revised manuscript text omitted]
 polarization of the electric field was perpendicular to the fibers and at a 12% humidity level by mass water.**

[Figure]

**Figure 4: B-scans showing the direct-air, direct and reflected waves measured over the sample, when the polarization of the electric field was perpendicular to the fibers and at a 12% humidity level by mass water.**

[Figure]

**Figure 5: Determination of the propagation velocity for the direct wave, from the arrival times. Both configurations with electric field parallel and perpendicular to the fibers are considered. In this case the humidity level was 18.18%. $R^2$ is Coefficient of determination of linear dependence.**

[Figure]

**Figure 6: Schematic view of the of the distances between antennas (blue line), thickness of the wood sample (green line) and the length of the reflected wave path (black line).**

[Figure]

5  **Figure 7: Variation of the dielectric constant with humidity from the direct wave method (WARR) for perpendicular and parallel polarization of E field.**

[Figure]

**Figure 8: The variation of the amplitude with distance from direct waves, for perpendicular and parallel polarization of E field (18.18% humidity).**

[Figure]

**Figure 9: Normalized amplitude of direct wave with respect to humidity, for perpendicular and parallel polarization of E field.**

**Table I: Dielectric constants in relation with humidity by mass water for direct and reflected waves.**

| Direct-wave (WARR) method | | | Reflected-wave method | | |
|---|---|---|---|---|---|
| Humidity (%) | E⊥ wood fibers $\varepsilon_r'$ | E// wood fibers $\varepsilon_r'$ | Humidity (%) | E⊥ wood fibers $\varepsilon_r'$ | E// wood fibers $\varepsilon_r'$ |
| 12 | 1.56 | 1.67 | 12 | 2.24 | 2.35 |
| 18.18 | 1.57 | 1.71 | 18.66 | 2.36 | 4.38 |
| 23.87 | 1.71 | 2 | 25.46 | 2.92 | 6.32 |
| 26.89 | 1.87 | 2.09 | 28.99 | 2.96 | 6.81 |
| 30.15 | 2.01 | 2.36 | 33.01 | 3.18 | 7.56 |
| 42.9 | 2.54 | 2.8 | 44.69 | 3.9 | 8.65 |
| 51.6 | 2.91 | - | 54.09 | 4.544 | 10.69 |
| 64.65 | 3.12 | - | 66.14 | 5.718 | 12.728 |